# Selective Upregulation of SIRT1 Expression in Retinal Ganglion Cells by AAV-Mediated Gene Delivery Increases Neuronal Cell Survival and Alleviates Axon Demyelination Associated with Optic Neuritis

**DOI:** 10.3390/biom12060830

**Published:** 2022-06-14

**Authors:** Ahmara G. Ross, Brahim Chaqour, Devin S. McDougald, Kimberly E. Dine, Thu T. Duong, Ryan E. Shindler, Jipeng Yue, Tehui Liu, Kenneth S. Shindler

**Affiliations:** 1Department of Ophthalmology, University of Pennsylvania, Philadelphia, PA 19104, USA; ahmara.ross@pennmedicine.upenn.edu (A.G.R.); brahim.chaqour@pennmedicine.upenn.edu (B.C.); devinmcdougald@gmail.com (D.S.M.); kdine@pennmedicine.upenn.edu (K.E.D.); thuduong@pennmedicine.upenn.edu (T.T.D.); shindler13@gmail.com (R.E.S.); jipeng.yue@pennmedicine.upenn.edu (J.Y.); thlindaliu@gmail.com (T.L.); 2F. M. Kirby Center for Molecular Ophthalmology, Scheie Eye Institute, University of Pennsylvania Perelman School of Medicine, Philadelphia, PA 19104, USA; 3Department of Neurology, University of Pennsylvania, Philadelphia, PA 19104, USA

**Keywords:** AAV7m8, optic neuritis, gene therapy, SIRT1, experimental autoimmune encephalomyelitis, inflammation, demyelination, optic nerve, retinal ganglion cells

## Abstract

Optic neuritis (ON), the most common ocular manifestation of multiple sclerosis, is an autoimmune inflammatory demyelinating disease also characterized by degeneration of retinal ganglion cells (RGCs) and their axons, which commonly leads to visual impairment despite attempted treatments. Although ON disease etiology is not known, changes in the redox system and exacerbated optic nerve inflammation play a major role in the pathogenesis of the disease. Silent information regulator 1 (sirtuin-1/SIRT1) is a ubiquitously expressed NAD^+^-dependent deacetylase, which functions to reduce/prevent both oxidative stress and inflammation in various tissues. Non-specific upregulation of SIRT1 by pharmacologic and genetic approaches attenuates RGC loss in experimental ON. Herein, we hypothesized that targeted expression of SIRT1 selectively in RGCs using an adeno-associated virus (AAV) vector as a delivery vehicle is an effective approach to reducing neurodegeneration and preserving vision in ON. We tested this hypothesis through intravitreal injection of AAV7m8.SNCG.SIRT1, an AAV2-derived vector optimized for highly efficient SIRT1 transgene transfer and protein expression into RGCs in mice with experimental autoimmune encephalomyelitis (EAE), a model of multiple sclerosis that recapitulates optic neuritis RGC loss and axon demyelination. Our data show that EAE mice injected with a control vehicle exhibit progressive alteration of visual function reflected by decreasing optokinetic response (OKR) scores, whereas comparatively, AAV7m8.SNCG.SIRT1-injected EAE mice maintain higher OKR scores, suggesting that SIRT1 reduces the visual deficit imparted by EAE. Consistent with this, RGC survival determined by immunolabeling is increased and axon demyelination is decreased in the AAV7m8.SNCG.SIRT1 RGC-injected group of EAE mice compared to the mouse EAE counterpart injected with a vehicle or with control vector AAV7m8.SNCG.eGFP. However, immune cell infiltration of the optic nerve is not significantly different among all EAE groups of mice injected with either vehicle or AAV7m8.SNCG.SIRT1. We conclude that despite minimally affecting the inflammatory response in the optic nerve, AAV7m8-mediated SIRT1 transfer into RGCs has a neuroprotective potential against RGC loss, axon demyelination and vison deficits associated with EAE. Together, these data suggest that SIRT1 exerts direct effects on RGC survival and function.

## 1. Introduction

Optic neuritis refers to an idiopathic inflammatory demyelinating disease of the optic nerve with a worldwide distribution and variable prognosis [1]. Typical and atypical features of ON include, among other symptoms, ocular pain with eye movement, unilateral vision loss, acute reduction in visual field size, color vision, central visual acuity, and afferent pupillary function [2]. ON is also the most common ocular and second most frequent manifestation of multiple sclerosis (MS) [3], which typically is characterized by unpredictable episodic neurological dysfunction caused by inflammatory demyelinating lesions in the optic nerve. Up to 75% of patients with ON convert to full-blown MS between 10 years and 15 years after clinical onset [4]. Although disease causes have not completely been identified, the underlying mechanism is suggested to likely involve infiltration of inflammatory cells (e.g., CD4^+^ T cells (known as T helper 17 cells) and CD8+ T cells) into and demyelination of optic nerve, axonal damage, and death of retinal ganglion cells (RGCs) [5], the projection neurons that convey visual signals from other retinal neurons to the brain. This neurodegenerative process leads to irreversible loss of vision. Current treatment options that are based on the use of immunosuppressants and anti-inflammatory drugs accelerate visual recovery after an acute episode of ON [6,7,8] but could neither have long-lasting effects nor rescue inflammatory demyelination of the optic nerve and RGC death, which are key interventional targets in many optic neuropathies [9].

Factors inducing RGC death and inflammatory demyelination of the optic nerve include excessive production and accumulation of damaging reactive oxygen species, defective axonal transport, trophic factor depletion and loss of RGC electrical activity [10,11]. Thus, effective treatment strategies must be designed to provide optimal conditions for mitigating the effects of these injurious stimuli to preempt dysfunction or restore healthy functional RGCs. Herein, we hypothesize that targeted expression of SIRT1 to RGCs rescues nerve fiber demyelination and RGCs death associated with ON.

SIRT1 is a ubiquitously expressed histone deacetylase that removes acetyl groups from lysine residues of histones, which results in chromatin compaction and transcription repression [12]. SIRT1 also targets non-histone proteins such as transcription factors and transcriptional coregulatory proteins and regulates their activities by altering their stability, activity, subcellular localization, DNA-binding ability, and protein–protein interactions. In the myeloid-specific SIRT1 knockout mouse [13,14], loss of SIRT1 function resulted in hyperacetylation/inactivation of NF-κB and increased expression of proinflammatory genes, which underscores the anti-inflammatory activities of SIRT1 [12]. In addition, SIRT1 was also shown to deacetylate several apoptosis-related proteins, including p53, Smad7, FOXO3 and FOXO4, and protect cells against damage-induced apoptosis [15,16,17].

Our laboratory has pioneered numerous studies demonstrating the neuroprotective potential of SIRT1 in optic neuropathy models [18,19]. Using the AAV2 vector as a delivery system, we showed that SIRT1 overexpression in multiple cell types in the retina driven by a ubiquitous promoter induced a small reduction in RGC loss in a model of experimental autoimmune encephalomyelitis (EAE) [20]. Only 21% of RGCs were transduced with this AAV2 vector, and it was unclear if SIRT1 expression in RGCs vs. SIRT1 expression in other retinal cells was important for promoting RGC survival. Similarly, pharmacological activation of SIRT1 using systemic activators to upregulate SIRT1 in all cells attenuated RGC loss in models of ON and traumatic optic nerve injury [19,21]. These studies further demonstrated that SIRT1 effects were, at least in part, associated with reduced oxidative stress.

The AAV2 serotype has been extensively used to transduce RGCs in the retina. However, RGC transduction with the native AAV2 vector varied extensively [22,23]. Here, we examined the neuroprotective effects of AAV-mediated SIRT1 expression selectively in RGCs in the mouse model of EAE. We used a synthetic AAV2-derived AAV7m8.SNCG.SIRT1 vector, which was designed to drive optimal human SIRT1 expression selectively in RGCs under the control of the gamma-synuclein (SNCG) promoter [24,25] and demonstrated potential protective effects from RGC trauma [25]. We found that intravitreal injection of AAV7m8.SNCG.SIRT1 vector reduced RGC loss and demyelination of the optic nerve and provided effective neuroprotection in the model of EAE.

## 2. Materials and Methods

### 2.1. Mice

We used female C57Bl/6J mice (the Jackson Laboratory, Bar Harbor, ME, USA). Mice were housed at the animal facility at the University of Pennsylvania in a 12-h light/dark cycle. Experiments were carried out according to the institutional ethical guidelines and following animal protocols approved by the Institutional IACUC and in compliance with federal regulations.

### 2.2. AAV7m8.SNCG.SIRT1 Design and Preparation

The AAV vectors AAV7m8.SNCG.eGFP and AAV7m8.SNCG.SIRT1 used in this study consist of the AAV7m8 capsid (a gift from John Flannery, UC-Berkeley, CA, USA), the human SNCG promoter [26], the cDNA encoding enhanced GFP (eGFP) protein or codon-optimized Human SIRT1 (transcript variant 1) cDNA (Origene, Rockville, MD, USA), and the bovine growth hormone (bGH) polyadenylation signal. Vectors were generated as previously described [23] and produced by the Research Vector Core at the Center for Advanced Retinal and Ocular Therapeutics (University of Pennsylvania, Philadelphia, PA, USA). Briefly, human codon-optimized *SIRT1* cDNA (GenScript, Piscataway, NJ, USA) was amplified and cloned into an AAV expression plasmid using the In-Fusion HD commercial cloning kit (Clontech Laboratories, Mountain View, CA, USA). The transgene was cloned downstream of the human SNCG promoter. The AAV expression cassettes were flanked by the AAV7m8 inverted terminal repeats. A pro-viral plasmid driving expression of eGFP was used and contains identical cis regulatory elements. All AAV vectors were purified with CsCl gradient.

### 2.3. Intravitreal Injections

Four-week-old mice were anesthetized by isoflurane inhalation, and their pupils were dilated by topical application of tropicamide 1% for 1 min, followed by application of phenylephrine 2.5%. A small incision was created with a 33½ gauge needle about 2 mm posterior to the limbus. A 10-μL Hamilton syringe attached to a 33-gauge blunt-end needle was inserted ~1.5 mm into the vitreous cavity. The location of the needle was visualized under a surgical microscope, and the needle was positioned directly above the optic nerve head with its opening oriented toward the temporal retina facing neither the optic nerve nor the lens. Two microliters of AAV vector preparation (1 × 10^10^ vector genomes) of either AAV7m8.SNCG.eGFP or AAV7m8.SNCG.SIRT1 were injected into each eye. The two eyes of each mouse received different injections (either vehicle alone, AAV7m8.SNCG.eGFP, or AAV7m8.SNCG.eGFP), allowing each eye to serve as an independent experimental endpoint. Vehicle-treated eyes were injected with an equivalent volume of vector dilution buffer (0.001% Pluronic F68 in PBS). Mice were kept warm while they were awakening from anesthesia.

### 2.4. EAE Induction and Scoring

Eight-week-old mice were immunized subcutaneously on the back with two injections of 150 μg myelin oligodendrocyte glycoprotein peptide (MOG_35–55_, GenScript, Piscataway, NJ, USA) emulsified in complete Freund’s adjuvant (CFA; Difco Laboratories, Franklin Lakes, NJ, USA) containing 2.5 mg/mL heat-killed *Mycobacterium tuberculosis*. Mice were further injected intraperitoneally with 200 ng *pertussis* toxin (List Biological, Campbell, CA, USA) in 0.1 mL of PBS at 0 and 48 h postimmunization with MOG_35–55_. Mice were weighed daily to ensure weight loss did not exceed 20% of the starting weight. To document disease induction, EAE clinical manifestations of progressive ascending paralysis were assessed daily using a standard scoring system: no disease = 0; partial tail paralysis = 0.5; tail paralysis or waddling gait = 1.0; partial tail paralysis and waddling gait = 1.5; tail paralysis and waddling gait = 2.0; partial limb paralysis = 2.5; paralysis of one limb = 3.0; paralysis of one limb and partial paralysis of another = 3.5; paralysis of two limbs = 4.0; moribund state = 4.5; death = 5.0 [27].

### 2.5. Optokinetic Response (OKR) Recording

The visual threshold was assessed by OKR recording using Optometry software and apparatus (Cerebral Mechanics Inc., Lethbridge, AB, Canada), as previously described [28]. Briefly, mice were surrounded by a virtual cylinder consisting of vertical lines rotating at varying frequencies, and their tracking behavior was assessed by an investigator blinded to mouse groups and treatment. Visual function is represented as the highest spatial frequency at which mice track a rotating cylinder consisting of 100% contrast grating. Because tracking is a temporal to nasal-specific reflex, counter-clockwise and clockwise rotations selectively test the right eye and the left eye, respectively.

### 2.6. Immunohistochemistry and Quantification of eGFP Expressing Cells and RGC Density

Mouse eyes were enucleated and fixed in 4% paraformaldehyde (PFA) overnight at 4 °C. Retinal cups were dissected, permeabilized and blocked in 2% Triton X-100, 10% normal donkey serum, and PBS. For RGC and hSIRT1staining, tissues were incubated at 4 °C with a rabbit anti-Brn3a antibody (Synaptic systems, Goettingen, Germany, Cat# 411 003) diluted 1:1000 and a goat anti-hSIRT1 antibody (ProSci, Poway, CA, USA, 42;428) used at 1:1000 dilution as well. Retinas were then washed three times and incubated with a secondary antibody solution containing donkey anti-rabbit AlexaFluor 488 and donkey anti-goat AlexaFluor 594 (1:1000 dilutions each) for one hour at room temperature. After several washes, retinas were flat-mounted onto glass slides with an aqueous mounting medium (Southern Biotech, Birmingham, AL, USA), covered with a coverslip and sealed. Immunofluorescence was detected using a Nikon Fluorescence microscope. Retinal images were taken with a 40× objective in 12 standard fields (1/6, 3/6, and 5/6 of the retinal radius from the center of the retina in each quadrant). Total RGCs counted in the 12 fields (0.034 mm^2^/field) per retinal sample, covering a total area of 0.407 mm^2^/retina, were recorded by an investigator masked to the experimental and control groups using ImageJ software (http://imagej.nih.gov/ij/ (accessed on 3 May 2022); provided in the public domain by the National Institutes of Health, Bethesda, MD, USA). Mice injected with the AAV7m8.SNCG.eGFP vector were stained with anti-Brn3a antibody only for RGC colocalization with an eGFP signal. The number of retinal eGFP^+^ and/or Brn3^+^ cells was determined as described above for the Brn3a+ SIRT1^+^ cells.

### 2.7. Optic Nerve Staining and Scoring

Histologic and immunohistochemical staining and scoring were performed as previously described [18,19,20,25]. Optic nerves were collected, fixed in 4% paraformaldehyde (PFA), and embedded in paraffin. Nerves were subsequently cut into 5-μm longitudinal sections and stained with hematoxylin and eosin (H&E). Inflammation was scored by grading the nerve section on a 0 to 4 point scale: no infiltration = 0; mild cellular infiltration = 1; moderate infiltration = 2; severe infiltration = 3; massive infiltration = 4. To assess myelination, sections were stained with luxol fast blue (LFB) and graded on a 0 to 3 point scale: 0 = no demyelination; 1 = scattered foci of demyelination; 2 = prominent foci of demyelination; and 3 = large (confluent) areas of demyelination. To assess RGC axon density, sections were stained with rabbit anti-neurofilament antibody 1:500 (Abcam, Cambridge, MA, USA) at 4 °C overnight followed by anti-rabbit secondary antibody (Vectastain Elite ABC Rabbit kit, Vector Laboratories, Newark, CA, USA) for 30 min at 37 °C. Avidin/Biotin Complex detection was performed by incubating with Vectastain Elite ABC reagent at 37 °C for 30 min and DAB (diaminobenzidine, Vector laboratories) substrate for 3 min. A photograph of the center of each longitudinal optic nerve section was taken at ×40 magnification by a masked investigator. Neurofilament staining optical density was quantified using ImageJ software (http://nih.gov, accessed on 3 June 2022).

### 2.8. Statistical Analyses

All data are represented as means ± SEM or means ± SD, as stated in the results. Differences between treatment groups with respect to OKR responses, RGC quantification, and optic nerve histopathology were compared using a one-way ANOVA followed by Tukey’s honest significant difference test using statistical software (GraphPad Prism 7.0; GraphPad Software, Inc., La Jolla, CA, USA). Differences were considered statistically significant at *p* < 0.05.

## 3. Results

### 3.1. AAV7m8.SNCG Efficiently and Selectively Transduces RGCs in the Mouse Retina

We used AAV7m8, a synthetic AAV2-derived vector redesigned through capsid and genome modification of the AAV2 capsid to enhance infection efficiency and tropism to retinal cells [23]. To increase AAV7m8 transduction efficiency to RGCs and achieve therapeutic effects in these cells, AAV7m8 was further modified to drive transgene expression by an RGC-selective SNCG promotor (Figure 1A). In preliminary experiments, we demonstrated the potential of AAV7m8.SNCG.eGFP, which carries SNCG promoter-driven eGFP to transduce RGCs in the mouse retina to a similar degree as in prior studies [25]. AAV7m8.SNCG.eGFP was injected intravitreally into 4-week-old mice, and the reporter transgene expression in RGCs was analyzed four weeks later (Figure 1B). The reporter expression profile was assessed by eGFP fluorescence combined with Brn3a labeling. Immunohistochemical staining of either flat mounts or cross-sections of retina showed that SNCG promoter-driven eGFP-positive cells were localized predominantly in the retinal ganglion cell layer and were selectively expressed in cells labeled with the Brn3a antibody (Figure 1C,D). In total, 90.54 ± 4.65% (mean ± SD) of eGFP-positive cells were also Brn3a-positive when counted in retinal cross-sections from three mice. eGFP fluorescence was only detected in injected eyes, not in contralateral eyes of the same mice. By quantifying the eGFP-positive cell population labeled with Brn3a antibody in retinal flat mounts, we found that AAV7m8.SNCG.eGFP delivered eGFP transgene to more than 55% of Brn3a-positive RGCs in the retina (Figure 1E). Similarly, when the AAV7m8 vector carrying the SIRT1 transgene was injected into the eye of wild-type mice, a similar selective transduction profile in RGCs was observed (Figure 1F). This vector achieved approximately 51% RGC transduction by quantifying the number of SIRT1-positive RGCs labeled with the Brn3a antibody (Figure 1G). Thus, the AAV7m8.SNCG vector efficiently and selectively transduced RGCs in the retina with two different transgenes.

### 3.2. AAV7m8-Mediated Delivery of the SIRT1 Gene Attenuates Visual Decline in the EAE Model

EAE mice were used to determine the effects of AAV7m8-mediated SIRT1 delivery on visual function and RGC survival during optic neuritis. Mild-moderate EAE disease was induced in 16 EAE mice (peak EAE score 1.818 ± 0.902 (mean ± SD); range 0.5–3.5). Eyes of four-week-old female C57Bl/6J mice were divided into four cohorts, with cohorts 1 and 4 receiving AAV7m8.SNCG.SIRT1 by intravitreal injection while cohorts 2 and 3 received vehicle and AAV7m8.SNCG.eGFP, respectively (cohort 1: AAV7m8.SNCG.SIRT1 (control), *n* = 13; cohort 2: vehicle (EAE), *n* = 7; cohort 3: AAV7m8.SNCG.eGFP (EAE), *n* = 8; cohort 4 AAV7m8.SNCG.SIRT1 (EAE), *n* = 17). Four weeks post-AAV injection, EAE was induced in cohorts 2, 3 and 4 through MOG_35–55_ immunization. We determined visual function in all four cohorts by OKR prior to EAE induction/MOG immunization and once every 7 days for up to 42 days postimmunization. Control (non-EAE) animals show unaltered visual acuity reflected by high OKR scores throughout the time course of the experiment, whereas visual acuity begins to decrease 14–21 days after EAE induction as reflected by lower OKR scores that eventually plateaued at 28 days and thereafter (Figure 2). OKR scores measured on days 28, 35 and 42 postimmunization were significantly reduced in sham (vehicle or AAV7m8.SNCG.eGFP) treated EAE groups when compared to those of the control non-EAE group (Figure 2). Interestingly, eyes that received AAV7m8.SNCG.SIRT1 largely maintained their vision and exhibited higher OKR scores than other EAE mouse eyes, with scores comparable to those of control animals, at multiple time points. Note that data obtained in control healthy mice administered with either AAV7m8.SNCG.SIRT1 or AAV7m8.SNCG.eGFP were similar; therefore, the results obtained in the latter group were not included in the presented data. In addition, a separate analysis of right and left eyes (data not shown) demonstrated similar average OKR scores and maintained statistical differences between treatment groups, suggesting no significant effects of within-subject inter-eye correlations.

### 3.3. AAV7m8.SNCG.SIRT1 Delivery Augments RGC Survival in EAE Mice

To assess the effects of SIRT1 delivery on neurodegeneration, retinas were harvested on day 42, flat-mounted and stained with antibodies against Brn3a. RGC density was determined as the sum of labeled RGCs counted in standardized areas of the retina. Representative immunofluorescence micrographs illustrating RGC density in retinal flat mounts around the optic nerve head from vehicle-, AAV7m8.SNCG.eGFP- or AAV7m8.SNCG.SIRT1-treated control and EAE mice are shown in Figure 3A. Quantitative analysis of RGC density showed a statistically significant decrease in RGC numbers in the EAE mouse cohorts sham-treated with vehicle or AAV7m8.SCNG.eGFP compared to non-EAE control mice (Figure 3B). AAV7m8-mediated expression of SIRT1 significantly improved RGC survival compared to the vehicle- and AAV7m8.SNCG.eGFP-treated cohorts of EAE mice. RGC axon density was assessed in optic nerve sections from *n* = 4 randomly selected eyes of each treatment cohort. The average optical density of neurofilament immunostaining was set as 100% in control, non-EAE mouse optic nerves (100 ± 6.16%, mean ± SD). Optic nerves from EAE mice sham-treated with vehicle (70.3 ± 2.50%) or AAV7m8.SNCG.eGFP (69.5 ± 6.20%) had a significant decrease in neurofilament staining (*p* < 0.05) as compared with controls, and EAE mice treated with AA7m8.SNCG.SIRT1 (80.1 ± 5.29%) had significantly more dense neurofilament staining than sham-vector-treated mice (*p* < 0.05).

### 3.4. AAV7m8-Mediated Expression of SIRT1 in RGCs Reduces Demyelination but Not Inflammation in the EAE Mouse Model

To determine the extent of inflammatory infiltrate in the optic nerve of treated and non-treated groups, longitudinal sections of the optic nerve were H&E stained and scored for immune cell infiltration. As shown in Figure 4A, optic nerves of both vehicle- and vector-treated EAE mice showed focal areas of cell infiltrate not present in non-EAE control mice. Quantitative analysis showed a significant increase in the relative level of inflammation in vehicle-, AAV7m8.SNCG.eGFP- and AAV7m8.SNCG.SIRT1-injected EAE mice as compared with non-EAE control mice treated with AAV7m8.SNCG.SIRT1, indicating that SIRT1 expression did not mitigate inflammation of the optic nerve in EAE mice (Figure 4B). Similarly, the effect of SIRT1 gene transfer on EAE-induced optic nerve demyelination was determined by staining optic nerve sections with LFB. Histological examination of LFB-stained sections from the control non-EAE group showed healthy nerve with no sign of demyelination, while those of vehicle- or AAV7m8-eGFP-injected EAE mice exhibited signs of optic nerve demyelination (Figure 5A). Statistical analysis of demyelination scores showed that optic nerve demyelination was significantly increased in EAE mice injected with either vehicle or AAV7m8.SNCG.eGFP when compared to non-EAE mice (Figure 5B). Conversely, demyelination scores were lower in AAV7m8.SNCG.SIRT1-injected EAE mice as compared with those of EAE mice injected with AAV7m8.SNCG.eGFP vector, indicating that selective SIRT1 expression in RGCs both increased RGC survival and protected against optic nerve demyelination.

## 4. Discussion

Gene therapy using AAVs as vehicles to deliver a functional transgene into cells is a promising strategy for the treatment of various ocular and non-ocular diseases. An important requirement for the therapeutic use of AAV-based gene delivery is to achieve effective, precise, and sustained transgene expression in targeted cells and minimize unwanted expression and effects in other cell types. The present study was undertaken to determine the effects of SIRT1 gain of function via a novel synthetic vector (AAV7m8) combined with an RGC-selective promoter (SNCG) on retina and optic nerve alterations and visual function in EAE-induced optic neuritis. The rationale for examining this SIRT1 delivery system is twofold. First, AAV7m8 was modified and optimized through capsid and genome design modification to increase its efficiency in delivering a therapeutic transgene compared to its previous parental AAV2 vector, the first FDA-approved AAV as a drug for the treatment of retinal diseases [29,30]. A study by Duong et al. has compared the transduction efficiency of 11 rAAV serotypes carrying a reporter gene and determined that AAV7m8 had superior transduction performances across in vitro and in vivo model systems with all vector concentrations tested [24]. This further supports the utility of this vector in preclinical and potentially clinical applications. Secondly, AAV7m8.SNCG.SIRT1, which drives transgene expression selectively in RGCs under the control of the SNCG promoter, was more efficient in transducing RGCs than reported in a prior study using a cytomegalovirus/chicken beta actin (CAG) hybrid promoter, which indiscriminately drives transgene expression in all cell types [25]. The use of the SNCG promoter allowed a highly efficient targeted transfer and translation of the functional transgene cassette into RGCs with greater than 90% cell selectivity. A previous study showed similar RGC-selective expression of eGFP using this construct in a traumatic injury model [25] although this study did not describe the expression pattern of the therapeutic human SIRT1 transgene. The selection of the SNCG promoter was derived from previous studies showing that the fraction of GCL cells expressing SNCG (45%) matched the estimated fraction of RGCs in the mouse GCL (41–44%), indicating that the SNCG promoter is active in nearly all RGCs [31]. In our study, the eGFP reporter gene driven by the SNCG promoter labeled Brn3a-positive cells but also some Brn3a-negative cells, likely because Brn3s are not expressed in all SNCG-positive RGCs [32]. The extent of RGC transduction may be underestimated, as other RGC markers such as RPBMS label a higher percentage of RGCs than Brn3a [33], with Brn3a used in the current studies for comparison to prior reports. Indeed, as shown in a recent study from our group, AAV7m8.SNCG.eGFP was twice more efficient than AAV2 with a constitutive promoter in transducing and expressing the transgene reporter in RGCs [25]. Here, we demonstrate that this promoter and capsid combination drives not only eGFP expression but also promotes hSIRT1 transduction and expression at a similar level.

With the AAV7m8.SNCG.SIRT1 tool in hand, we were able to perform a proof-of-concept study testing whether RGC-targeted, SIRT1 expression is a potential gene therapy for optic neuritis. We used the EAE model, which is characterized by an immune reaction that causes RGC loss and altered visual function [34]. Through intravitreal injection of the AAV7m8.SNCG.SIRT1 vector, we found that SIRT1 delivery predominantly to RGCs significantly reduced visual deficits that accompany EAE onset and progression as determined by OKR response. Parallel to the preservation of visual function, SIRT1 decreased optic nerve demyelination and reduced RGC loss. The degree of RGC neuroprotection was particularly notable given that only a little more than half of Brn3a positive RGCs were transduced with SIRT1, although additional Brn3a negative RGCs may also have been transduced, and it is possible that surviving RGCs secrete other local factors that may contribute to the survival of neighboring non-transduced RGCs. Interestingly, even though RGC death is a secondary effect of optic nerve inflammation, the inflammation scores were not significantly affected by SIRT1 expression in EAE mice. Results are similar to prior studies using pharmacologic or non-cell-specific gene therapy methods to upregulate SIRT1, which consistently demonstrated RGC neuroprotection in optic neuritis without suppression of optic nerve inflammation [18,19,20,27]. Other studies have further documented the neuroprotective role of SIRT1 in other optic nerve disease models. Neuroprotection by SIRT1 activators has been reported in optic neuritis models simulated in a neurotropic strain of mouse hepatitis virus, MHV-A59 [35]. In the optic nerve crush model, wherein RGC injury is induced by direct trauma, SIRT1 activation or overexpression delayed and reduced RGC loss and traumatic optic nerve damage, at least in part by reducing reactive oxygen species levels and increasing mitochondrial metabolism [21,25]. Our group also showed that pharmacological activation of SIRT1 provided RGC neuroprotective effects in primary retinal cell cultures under oxidative stress conditions [36]. All of these prior studies support a neuroprotective role of SIRT1 on RGCs, although most of them have used non-RGC-targeted strategies to upregulate SIRT1. For EAE optic neuritis studies specifically, it was not clear whether SIRT1 upregulation was needed directly in RGCs, or whether upregulation in other cells induced secondary effects leading to RGC survival. Results here indicate that SIRT1 expression directly within RGCs is indeed sufficient to reduce RGC loss.

Even though SIRT1 exerts anti-inflammation-independent neuroprotective effects in our optic neuritis model, other studies reported that SIRT1 regulates genes and proteins involved not only in apoptosis and senescence but also inflammation pathways in different biological settings and tissues [32]. SIRT1 deficiency in mouse microglia activates IL-1 transcription through hypomethylation of specific CpG sites on IL-1 proximal promoter [37]. In addition, a study by Ye et al. [38] suggested that expression of SIRT1 in BV2 microglial cells decreased the expression of inflammatory factors such as TNF-α and IL-6, which indicates a direct effect of SIRT1 on proinflammatory cytokine expression during microglial activation [37]. Mechanistically, SIRT1 deacetylase activity suppresses inflammation through interaction and deacetylation of the RelA/p65 subunit of NFκB, which inhibits its transactivation potential and the transcription of NFκB-dependent inflammatory cytokines [39]. Similarly, SIRT1 inhibits High-Mobility Group Box 1 (HMGB1), which, if and when released in the extracellular milieu, activates inflammatory cells such as macrophages [40,41]. Thus, the anti-inflammatory effect of SIRT1 could largely be attributed to SIRT1-dependent repression of NFκB and HMGB1 signaling. A potential explanation for the lack of correlation between therapies that upregulate SIRT1 expression and inflammation scores in EAE mice is that the strategy used to target SIRT1 to RGCs in the current study limited its effects predominantly to these cells. Because of the intracellular localization of SIRT1 and its binding partners/substrates, inflammatory cells remained out of reach to SIRT1 expression, and the effects of SIRT1 were limited to the RGCs that produce it. However, our previous studies have shown that ubiquitous, AAV2-mediated overexpression of SIRT1 also failed to reduce optic nerve inflammation in EAE [20,28], although the extent of SIRT1 expression in retinal inflammatory cells is unknown. Of note, pharmacologic upregulation of SIRT1 has also consistently shown an ability to attenuate RGC loss without suppressing optic nerve inflammation in EAE [18,19,20,27], suggesting that inflammatory responses specifically in EAE optic neuritis may not be modulated by SIRT1. Interestingly, prior studies using other strategies to upregulate SIRT1 have had variable effects on demyelination in EAE optic neuritis [10,18,27] unlike the significant reduction found here with RGC-selective gene delivery. Given that inflammatory demyelination is the likely cause of RGC loss in EAE, the significantly improved RGC survival in SIRT1-treated mice on day 42 postimmunization may have provided a better protective effect against optic nerve degeneration and/or facilitated rapid remyelination of optic nerve axons following EAE. Indeed, remyelination is recognized to occur following acute inflammatory exacerbations in demyelinating disease, suggesting future studies may be warranted to examine whether increased myelination induced by SIRT1 gene delivery is due to the prevention of primary myelin degeneration or due to remyelination of surviving axons. This SIRT1 effect could bolster the therapeutic usefulness of AAV-mediated SIRT1 expression for the treatment of demyelinating and neurodegenerative diseases at large [20,25,28].

Other groups have shown similar potential for SIRT1-mediated neuroprotection. Direct activation of SIRT1 under pathological conditions or natural products that can contribute to the attenuation of oxidative stress by targeting SIRT1 were shown to increase RGC survival and preserve visual acuity [42,43]. Oxidative stress alters the expression of several genes such as nuclear factor E2 related factor (Nrf2), nuclear factor E2 related factor 2 (Nef2), nuclear factor kappa B (NF-κB), pancreatic and duodenal homeobox factor 1 (PDX1), and forkhead box class O (FOXO) [12]. It has been shown that SIRT1 forms a complex with these molecules and positively regulates their expression, thus providing protective effects against oxidative stressors in different organs. Consistent with this, AAV-mediated expression of Nrf2 reduced RGC death and induced SIRT1-like effects in the model of EAE [20]. Mechanistically, SIRT1 induces expression and activation of Nrf2, which under oxidative stress conditions, translocates into the nucleus and interacts with antioxidant response elements, promoting the expression of cytoprotective target genes such as glutathione-S-transferases, NAD(P)H: quinone oxidoreductase (NQO1), NQO2, γ-glutamyl cysteine synthase, glucuronosyltransferase, ferritin, and heme oxygenase-1 (HO-1) [44].

## 5. Conclusions

RGC survival and optic nerve myelination can safely be preserved through intravitreal injection of AAV7m8-mediated SIRT1 expression using the SNCG promoter. This therapeutic approach works independently of the autoimmune inflammatory reaction associated with EAE. Furthermore, results demonstrate that SIRT1 overexpression selectively in RGCs recapitulates neuroprotective effects induced by broader therapeutic strategies that increase SIRT1 in multiple cell types, demonstrating that direct SIRT1 signaling within RGCs is critical. Our results underscore the therapeutic utility of the RGC-selective strategy to express SIRT1 to provide effective neuroprotection while limiting the potential for off-target effects in other cells.

## Figures and Tables

**Figure 1 biomolecules-12-00830-f001:**
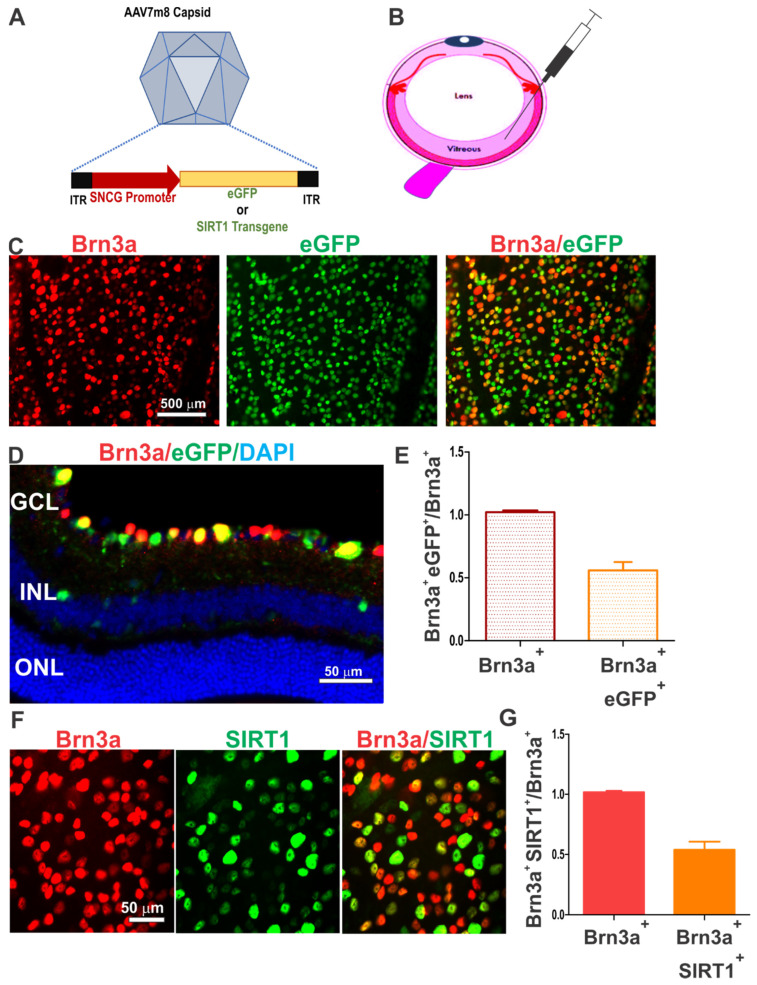
AAV7m8 transduced RGCs with high efficiency after single intravitreal injection. (**A**) Schematic representation of the AAV7m8 capsid and genome structure for enhanced transduction and RGC selective targeting. AAV7m8 was modified to drive the expression of the transgene (e.g., eGFP, SIRT1) under the control of the SNCG promoter. Transgene is placed between 2 internal terminal repeats (ITRs). (**B**) AAV7m8 administration is achieved through single injection of the AAV vector into the vitreous humor of the mouse eye. (**C**) Fluorescence micrographs of flat mounted preparation of mouse retina after intravitreal injection of the AAV7m8.SNCG.eGFP vector. RGCs were labeled with a monoclonal antibody against Brn3a. Merged eGFP (green) and Brn3a (red) signal is shown. (**D**) Representative fluorescence micrographs of retinal cross-section after intravitreal administration of AAV7m8.SNCG.eGFP and immunostaining with Brn3a antibody. Note that cells expressing SNCG promoter-driven eGFP are localized mainly in the ganglion cell layer (GCL). Cells within the inner nuclear layer (INL) and outer nuclear layer (ONL) were not transduced with the AAV vector. (**E**) Quantification of the population of RGCs transduced with the AAV7m8.SNCG.eGFP vector. Data shown are means ± SEM (*n* = 4). (**F**) Representative fluorescence micrographs of retinal flat mounts after intravitreal administration of AAV7m8.SNCG.SIRT1 and dual immunostaining with antibodies against Brn3a and human SIRT1. (**G**) Quantification of the population of SIRT1^+^ RGCs labeled with the Brn3a antibody in the retina of mice injected with the AAV7m8.SNCG.SIRT1 vector. Data shown are means ± SEM (*n* = 6).

**Figure 2 biomolecules-12-00830-f002:**
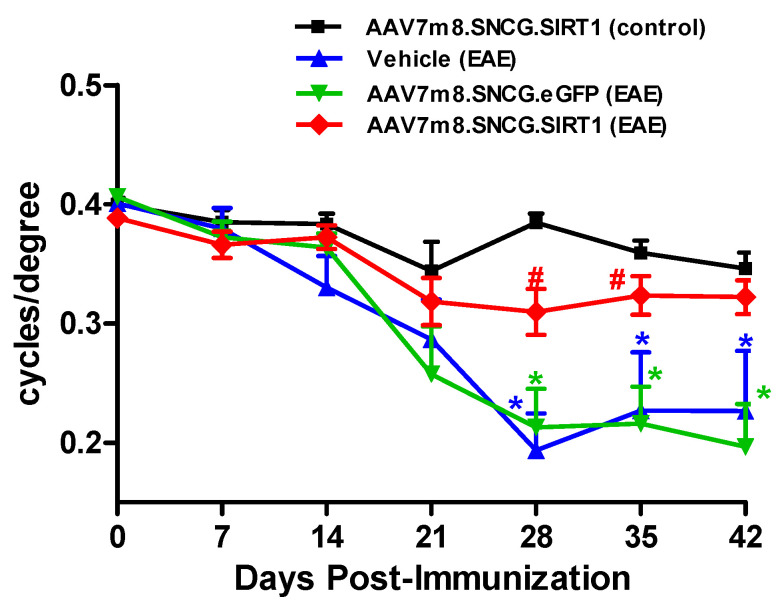
Effects of AAV7m8-mediated selective expression of SIRT1 in RGCs on visual function. Visual function was determined by OKR tracking using the Optometry apparatus and software. Measurements were performed prior to MOG immunization and once every 7 days for up to 42 days postimmunization in AAV7m8.SNCG.eGFP and AAV7m8.SNCG.SIRT1-treated and non-treated control and EAE mice. Data are presented as means ± SEM. * *p* < 0.05, AAV7m8. SNCG.SIRT1 (control) versus either vehicle (EAE) or AAV7m8.SNCG.eGFP (EAE), and ^#^ *p* < 0.05, AAV7m8. SNCG.SIRT1 (EAE) versus AAV7m8. SNCG.eGFP. Statistical significance was determined at each time point by one-way ANOVA with Tukey’s multiple comparisons test (EAE) (*n* = 13 for AAV7m8.SNCG.SIRT1 (control); *n* = 7 for vehicle (EAE); *n* = 8 for AAV7m8.SNCG.eGFP (EAE); *n* = 17 for AAV7m8.SNCG.SIRT1).

**Figure 3 biomolecules-12-00830-f003:**
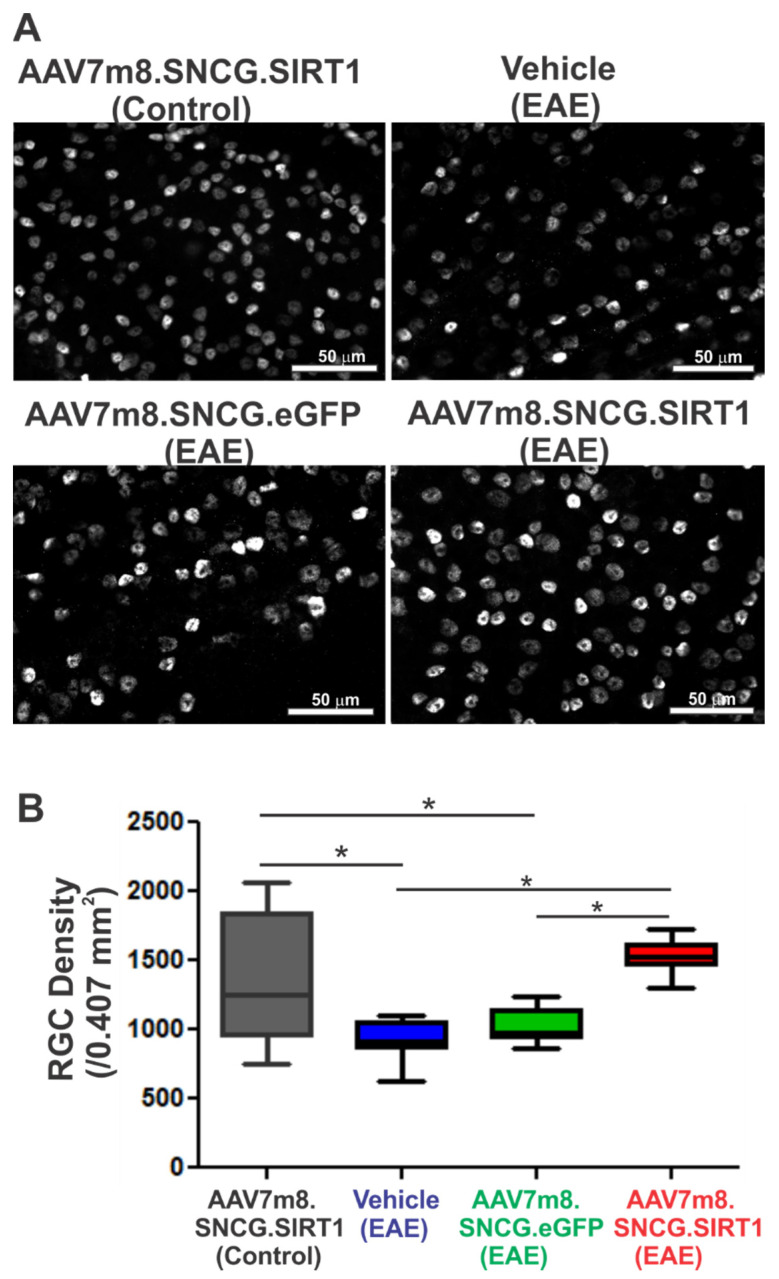
Effects of AAV7m8-mediated selective expression of SIRT1 in RGCs on RGC survival. (**A**) Representative immunofluorescence micrographs of RGC staining in the central retina of vehicle-, AAV7m8.SNCG.eGFP- and AAV7m8.SNCG.SIRT1-treated control and EAE mice. Scale bar, 50 μm. (**B**) The total number of labeled RGCs present in 12 standardized retinal fields was determined. The average number of surviving RGCs per total sampled area of retina of control and EAE mice is shown in the graph. Values are means ± SEM. * *p* < 0.05 by one-way ANOVA and Tukey’s multiple comparisons test (*n* = 13 for AAV7m8.SNCG.SIRT1 (control); *n* = 7 for vehicle (EAE); *n* = 8 for AAV7m8.SNCG.eGFP (EAE); *n* = 17 for AAV7m8.SNCG.SIRT1).

**Figure 4 biomolecules-12-00830-f004:**
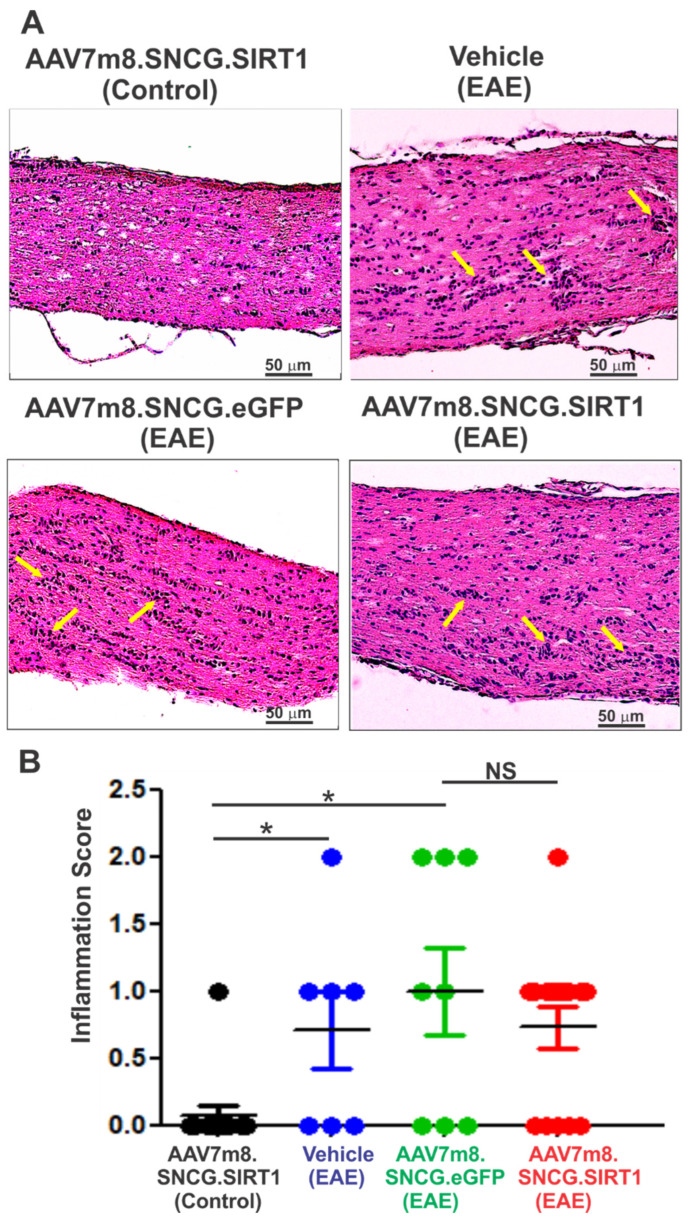
Effects of AAV7m8-mediated SIRT1 gene transfer in RGCs on optic nerve inflammation. (**A**) H&E staining of optic nerve longitudinal sections from vehicle-, AAV7m8.SNCG.eGFP- or AAV7m8.SNCG.SIRT1-treated mice are shown. Yellow arrows indicate small foci of inflammatory cell infiltrates in representative optic nerve photos. Optic nerve sections were imaged with 20× objective lens. Scale bar, 50 μm. (**B**) Inflammation scores determined by ratings of the number of infiltrating cell clusters. Data represented as mean ± SEM. * *p* < 0.05 by one-way ANOVA and Tukey’s multiple comparisons test (*n* = 13 for AAV7m8.SNCG.SIRT1 (control); *n* = 7 for vehicle (EAE); *n* = 8 for AAV7m8.SNCG.eGFP (EAE); *n* = 17 for AAV7m8.SNCG.SIRT1). NS = not significant.

**Figure 5 biomolecules-12-00830-f005:**
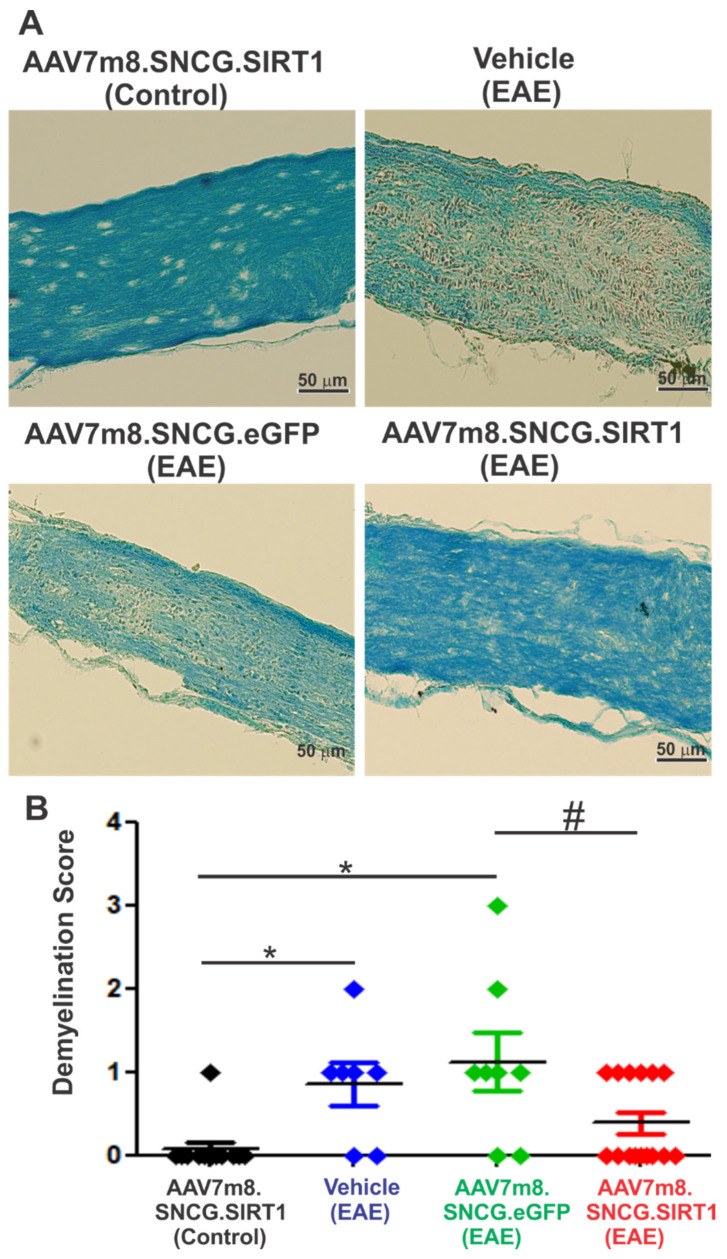
Effects of AAV7m8-mediated SIRT1 gene transfer in RGCs on optic nerve demyelination. (**A**) Representative photos of optic nerve longitudinal sections showing LFB staining. Optic nerve sections were visualized with light microscopy with 20× objective lens. Scale bar, 50 μm. (**B**) LFB stained areas of optic nerve were quantified. Data represented as mean ± SEM. * *p* < 0.05 versus AAV7m8.SNCG.SIRT1-treated non-EAE control mice. ^#^ *p* < 0.05 versus AAV7m8.SNCG.eGFP sham-treated EAE mice by one-way ANOVA and Tukey’s HSD post-test (*n* = 13 for AAV7m8.SNCG.SIRT1 (control); *n* = 7 for vehicle (EAE); *n* = 8 for AAV7m8.SNCG.eGFP (EAE); *n* = 17 for AAV7m8.SNCG.SIRT1).

## Data Availability

All data are available within the manuscript or upon request to the corresponding author.

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
