# Peer review of "Selective Upregulation of SIRT1 Expression in Retinal Ganglion Cells by AAV-Mediated Gene Delivery Increases Neuronal Cell Survival and Alleviates Axon Demyelination Associated with Optic Neuritis"

_biomolecules, 2022, doi:10.3390/biom12060830_

Round 1

Reviewer 1 Report

This is a manuscript reports the neuroprotective effect of AAV-mediated SIRT1 overexpression in optic neuritis/EAE mouse model. Similar neuroprotective effect has been report by the same group in optic nerve crush model before. Here in this study, the same AAV2 capsid variant, AAV7m8 was used to delivery SIRT1 in mouse RGCs after intravitreal injection. Visual function assay OKR was used to evaluate visual acuity of the EAE mice with or without SIRT1 overexpression, and RGC density, inflammation cell infiltration in optic nerve, and optic nerve myelination were examined to demonstrate the protective effect. The result is interesting by providing potential gene therapy strategy for neuroprotection in optic neuropathies. However, there are some concerns listed below that needs to improve the quality and interpretation of the manuscript.

Major points:

  1. RGC marker: although Brna3a has been used historically as RGC marker, it has been shown that RBPMS is a more reliable RGC marker. Brn3a at most only label 80% RBPMS+ RGC (Rodriguez et al., 2014) and Brn3a express is decreased dramatically in diseases, even live RGCs.
  2. AAV7m8 was developed for photoreceptor targeting through its superb penetrating capability. However, the specific RGC targeting by AAV7m8 is not definitively established. In Fig. 1C, D, clearly AAV7m8-human SNCG-mediated GFP expression is not restricted to GCL layer, nor Brn3a+ RGCs. It is ok to use it after intravitreal injection to infect RGCs, but it will need much more comprehensive study if they want to claim AAV7m8 or human SNCG promoter can achieve RGC specific targeting.  
  3. SNCG promoter: the Chaffiol A, Mol Ther 2017 paper they cited is a human SNCG promoter, it should be clearly indicated as it won’t confused with mouse SNCG promoter. The mouse SNCG promoter has been shown to be much more potent than human SNCG promoter in another study, both in mouse RGCs in vivo and human iPSC-derived human RGCs, Wang Q, et al J of Neuroscience 2020, and the sequence of human SNCG promoter is very different to the mouse SNCG promoter.
  4. 2: visual acuity measurement by OKR is a good visual function assay, however, it also depends on the body/head movement of the mice, which is significantly affected by EAE, especially at late stage, increasing the inconsistency of this assay in EAE mice. The data would be significantly strengthened if an additional visual function assay can be used, such as VEP and PERG, which can be tested without the need of the animal movement.
  5. 2: The clinic score of the testing mice should be presented to demonstrate the EAE model.
  6. 3: The quantification Fig. 3B showed that SIRT1 treated group has similar or even slight better RGC density than control group (without EAE), so the 51% RGC transduction rate from Fig. 1G achieved 100% protection? Maybe RBPMS staining will show the true RGC survival. The number of animals in each group should be listed in the figure legend.
  7. How about RGC axon survival in the optic nerve?
  8. 4/5: The number of animals in each group should be listed in the figure legend.

Author Response

Please find a point-by-point response to the Reviewer’s comments here:

Reviewer 1

  1. RGC marker: although Brna3a has been used historically as RGC marker, it has been shown that RBPMS is a more reliable RGC marker. Brn3a at most only label 80% RBPMS+ RGC (Rodriguez et al., 2014) and Brn3a express is decreased dramatically in diseases, even live RGCs.

Response: We thank the Reviewer for raising this important point, and agree there are advantages and limitations of using different RGC markers. Due to the limited time the editors allow for revisions, it would not be feasible to repeat this experiment and use RPBMS staining.  Of note, we elected to still use Brn3a as this provides important perspective when interpreting the current study in the context of our prior publications examining the effects of pharmacologic and non-cell-selective gene delivery approaches to upregulate SIRT1 in optic neuritis. While Brn3a exhibits limitations as a cell marker, this limitation if anything suggests that we have likely undercalculated the percentage of RGCs transduced with our therapeutic vector and may explain why the vector is found in some Brn3a negative cells, and may also suggest how the vector could be so effective.  We have added discussion of this in the revised manuscript. Future studies will be considered to examine expression levels in RPBMS+ RGCs.

  1. AAV7m8 was developed for photoreceptor targeting through its superb penetrating capability. However, the specific RGC targeting by AAV7m8 is not definitively established. In Fig. 1C, D, clearly AAV7m8-human SNCG-mediated GFP expression is not restricted to GCL layer, nor Brn3a+ RGCs. It is ok to use it after intravitreal injection to infect RGCs, but it will need much more comprehensive study if they want to claim AAV7m8 or human SNCG promoter can achieve RGC specific targeting.

Response: We agree we have not demonstrated that the current strategy is fully RGC-specific, and regret it was not more clear that we were not claiming this.  We were careful to refer to this as a RGC-selective strategy, not specific, based on the high percentage of RGCs transduced.  Indeed, AAV7m8 was developed for photoreceptor targeting following subretinal injection, and its use is being adapted to target the inner most retinal layers following intravitreal injection.  We have added additional data showing that this construct and route of administration led to selective RGC expression with greater than 90% of cells expressing transgene in retinal cross-sections being brn3a positive.

  1. SNCG promoter: the Chaffiol A, Mol Ther 2017 paper they cited is a human SNCG promoter, it should be clearly indicated as it won’t confused with mouse SNCG promoter. The mouse SNCG promoter has been shown to be much more potent than human SNCG promoter in another study, both in mouse RGCs in vivo and human iPSC-derived human RGCs, Wang Q, et al J of Neuroscience 2020, and the sequence of human SNCG promoter is very different to the mouse SNCG promoter.

Response: We cite the Chaffiol paper to indicate we used the same human SNCG promoter.  We have specified this in the methods to make it more clear.

  1. 2: visual acuity measurement by OKR is a good visual function assay, however, it also depends on the body/head movement of the mice, which is significantly affected by EAE, especially at late stage, increasing the inconsistency of this assay in EAE mice. The data would be significantly strengthened if an additional visual function assay can be used, such as VEP and PERG, which can be tested without the need of the animal movement.

Response: Mice developed moderate EAE disease (scores have been added; see responses below) with ascending paralysis not reaching the cervical spine; thus head movements were not affected and not expected to significantly alter testing.  We have obtained reliable and reproduceable OKR scores in such moderate EAE mice over many experiments.  While we agree obtaining multiple functional tests could be reassuring, the advantages would be weighed against subjecting mice to additional stress which itself is known to reduce EAE severity.  In addition, the time allotted by the editors for these revisions would not allow us to repeat the experiments with additional functional testing, but these are outcomes to be considered in future studies.

  1. 2: The clinic score of the testing mice should be presented to demonstrate the EAE model.

Response: The mean peak EAE score, and range of scores across all mice has been added, demonstrating mild-moderate disease induction.

  1. 3: The quantification Fig. 3B showed that SIRT1 treated group has similar or even slight better RGC density than control group (without EAE), so the 51% RGC transduction rate from Fig. 1G achieved 100% protection? Maybe RBPMS staining will show the true RGC survival. The number of animals in each group should be listed in the figure legend.

Response: While statistical analysis shows no difference between the control group and the SIRT1 treated group, we would not claim there was 100% protection.  Re-graphing of the data using a box and whisker plot as suggested by Reviewer 3 shows there were a couple of outliers in the control the group but overall there is not likely 100% protection.  Nonetheless, as the Reviewer noted the degree of protection is still potentially surprising for 51% transduction.  We added discussion about potential explanations including how the use of Brn3a as our marker likely means we have under-estimated the number of transduced RGCs, as well as the possibility that transduced RGCs may produce factors that help support neighbouring cells, something that may warrant further investigation in the future. Numbers for each group have been added to the figure legend.

  1. How about RGC axon survival in the optic nerve?

Response: We have added new data measuring the optical density of neurofilament staining of optic nerves.  As in prior studies, this indirect measure of axon survival shows significant improvement in treated EAE mice. Direct axon counting of optic nerve cross-sections was not possible as longitudinal optic nerve sections were cut to best visualize and quantify optic nerve inflammation and demyelination.

  1. 4/5: The number of animals in each group should be listed in the figure legend.

Response: Numbers for each group have been added to the figure legends.

Reviewer 2 Report

The authors used  EAE model mice with an injection of AAV-mediated gene into the vitreous cavity to express SIRT-1 in RGCs, and analyzed visual function, RGC counts, and the degree of inflammation. The results showed that visual function and RGC counts in the treatment group were statistically significantly better than those in the control group, even though there was no statistically significant differences between the two groups in the degree of inflammation. These results are very interesting in terms of the neuroprotective effect of SIRT-1. However, an additional explanation is needed regarding one point below.   The authors wrote that the two  eyes of each mouse received different injections (either vehicle alone, AAV7m8.SNCG.eGFP (→SIRT-1?), or AAV7m8.SNCG.eGFP) allowing each eye to serve as an independent experimental end point in the materials and methods section. According to the mentioned above, did different cohorts include different eyes of a same mouse? If so, the results might be affected by the injection of the fellow eye (perhaps through general circulation).

Author Response

Please find a point-by-point response to the Reviewer’s comments here:

Reviewer 2                                                                                                                                           

Comment 1:   The authors wrote that the two  eyes of each mouse received different injections (either vehicle alone, AAV7m8.SNCG.eGFP (→SIRT-1?), or AAV7m8.SNCG.eGFP) allowing each eye to serve as an independent experimental end point in the materials and methods section. According to the mentioned above, did different cohorts include different eyes of a same mouse? If so, the results might be affected by the injection of the fellow eye (perhaps through general circulation).

Response: We thank the Reviewer for raising this important concern.  This was not observed, as mice injected with eGFP constructs in one eye did not show detectable eGFP expression in the contralateral eye. This is now stated in the results. While low levels of vector, below the threshold of detectable transgene, still might possibly have done this, such low levels are unlikely to affect results; and even if they do, then this would potentially serve to strengthen the observed treatment effect as sham treated EAE eyes might have partial protection from low levels of therapeutic vector.

Reviewer 3 Report

In this interesting manuscript, Ross and colleagues examined the effects SIRT1 expression, selectively in RGCs using AAV vector in mice with experimental autoimmune encephalomyelitis (EAE). They found that SIRT1-vector injected EAE mice maintain better visual acuity, assessed by optokinetic response (OKR). Additionally, RGC loss and axon demyelination was reduced in animals with increased SIRT1 expression. Immune cell infiltration was however, unchanged in treated and untreated EAE mice. They concluded, that SIRT1 exerts direct effects on RGC survival and function.

The study adds to the literature corroborating and extending on previously reported results. However, I do have some points of concern that need to be addressed:

1.) General statistics: If both eyes were examined by OKR and both optic nerves/retinae from one animal were included in the analysis, the authors should perform a statistical test accounting for the within-subject inter-eye/optic nerve correlations, e.g. a linear mixed effects model. It is inadequate to enter both eyes/optic nerves of the subjects as independent datapoints into the analysis as they are statistically dependent.

2.) The authors show nicely the improved RGC survival and reduced demyelination in the ON after vector injection. However, the intracellular mechanisms remain mainly unexplained. Especially the fact, that targeted SIRT1 expression in the RGCs leads to a reduced demyelination in the ON should be further assessed or at least discussed more extensively.  

The authors could also elucidate downstream elements of SIRT1/Nrf2 involved in antioxidative effects discussed by the authors, such as glutathione-S-transferases, NQO1/2, γ-glutamyl cysteine synthase, glucuronosyltransferase, ferritin and/or HO-1 by qPCR or western blot in the optic nerve and the retina.

3.) Continuously distributed data should be displayed either by showing all data points in the bar graphs or by using box-and-whisker plots, with all elements to improve transparency of the data.

4.) Why are the timepoints 28, 35 and 42 shown again in bar graphs? The differences can be seen and significant differences can also be marked in the x-y graph (Figure 2A)

5.) Figure quality: Figure 1F sems to be squeezed and the scale bar is hard to read. The x-axis legend of Figure 1G interferes with the axis. In Figure 2A, the letter “A” is partly hidden. The legend of Figure 2B-D looks stretched. In Figure 5B, the colours of the bar graphs are different to the other figures. Please revise all figures for better quality and inconsistencies.

Minor:

-The EAE score of the animals could be provided.  

Author Response

Please find a point-by-point response to the Reviewer’s comments here:

Reviewer 3

  • General statistics: If both eyes were examined by OKR and both optic nerves/retinae from one animal were included in the analysis, the authors should perform a statistical test accounting for the within-subject inter-eye/optic nerve correlations, e.g. a linear mixed effects model. It is inadequate to enter both eyes/optic nerves of the subjects as independent datapoints into the analysis as they are statistically dependent.

Response: We thank the Reviewer for raising this important concern.  While we have shown in the past that the variable nature of the optic neuritis disease in our animals limits inter-eye correlations and have used eyes with different treatments independently, re-confirming this here strengthens the results.  We took advantage of our study design whereby animals received different treatments in the right and left eyes, and these were randomly distributed in terms of whether or not the right eye received the treatment (SIRT1) vector or a sham treatment.  We re-analysed the data using only the right eyes of each mouse and found that despite only having half the power, differences between groups remained significant.  Thus, we state in the results that it appears there is little effect of inter-eye correlations.  Specific data (mean ± SD) are listed here:

  • The authors show nicely the improved RGC survival and reduced demyelination in the ON after vector injection. However, the intracellular mechanisms remain mainly unexplained. Especially the fact, that targeted SIRT1 expression in the RGCs leads to a reduced demyelination in the ON should be further assessed or at least discussed more extensively.  The authors could also elucidate downstream elements of SIRT1/Nrf2 involved in antioxidative effects discussed by the authors, such as glutathione-S-transferases, NQO1/2, γ-glutamyl cysteine synthase, glucuronosyltransferase, ferritin and/or HO-1 by qPCR or western blot in the optic nerve and the retina.

Response: We thank the Reviewer for raising these important points. Some of these downstream mechanisms have been examined in prior studies as cited in the current paper, and the current study focused on a new therapeutic approach and outcome measures.  As suggested, we expanded our discussion about the observed improvement in demyelination.

  • Continuously distributed data should be displayed either by showing all data points in the bar graphs or by using box-and-whisker plots, with all elements to improve transparency of the data.

Response: We have replaced the bar graphs in figures 3, 4 and 5 as suggested.

  • Why are the timepoints 28, 35 and 42 shown again in bar graphs? The differences can be seen and significant differences can also be marked in the x-y graph (Figure 2A)

Response: We showed these time points separately simply to highlight the significant differences.  We have removed the bar graphs in the revised manuscript and now marked the differences in the x-y graph as suggested.

  • Figure quality: Figure 1F sems to be squeezed and the scale bar is hard to read. The x-axis legend of Figure 1G interferes with the axis. In Figure 2A, the letter “A” is partly hidden. The legend of Figure 2B-D looks stretched. In Figure 5B, the colours of the bar graphs are different to the other figures. Please revise all figures for better quality and inconsistencies.

Response: We have revised all figures as suggested.

Minor:

-The EAE score of the animals could be provided.  

Response: The mean peak EAE score, and range of scores across all mice has been added, demonstrating mild-moderate disease induction.

Round 2

Reviewer 1 Report

Most questions have been addressed and due to the time limitation, other questions have been explained adequately. 

Reviewer 3 Report

Thank you, all points raised were addressed.